# A Description of the Yield of Genetic Reinvestigation in Patients with Inherited Retinal Dystrophies and Previous Inconclusive Genetic Testing

**DOI:** 10.3390/genes14071413

**Published:** 2023-07-08

**Authors:** Maria Areblom, Sten Kjellström, Sten Andréasson, Anders Öhberg, Lotta Gränse, Ulrika Kjellström

**Affiliations:** 1Ophthalmology, Department of Clinical Sciences Lund, Lund University, Skane University Hospital, 221 85 Lund, Sweden; 2S:t Erik Eye Hospital, 171 64 Solna, Sweden; 3Novartis Sverige AB, 164 40 Stockholm, Sweden

**Keywords:** inherited retinal dystrophy, next generation sequencing, DNA analysis, phenotype–genotype correlation, re-analysis

## Abstract

In the present era of evolving gene-based therapies for inherited retinal dystrophies (IRDs), it has become increasingly important to verify the genotype in every case, to identify all subjects eligible for treatment. Moreover, combined insight concerning phenotypes and genotypes is crucial for improved understanding of thevisual impairment, prognosis, and inheritance. The objective of this study was to investigate to what extent renewed comprehensive genetic testing of patients diagnosed with IRD but with previously inconclusive DNA test results can verify the genotype, if confirmation of the genotype has an impact on the understanding of the clinical picture, and, to describe the genetic spectrum encountered in a Swedish IRD cohort. The study included 279 patients from the retinitis pigmentosa research registry (comprising diagnosis within the whole IRD spectrum), hosted at the Department of Ophthalmology, Skåne University hospital, Sweden. The phenotypes had already been evaluated with electrophysiology and other clinical tests, e.g., visual acuity, Goldmann perimetry, and fundus imaging at the first visit, sometime between 1988–2015 and the previous—in many cases, multiple—genetic testing, performed between 1995 and 2020 had been inconclusive. All patients were aged 0–25 years at the time of their first visit. Renewed genetic testing was performed using a next generation sequencing (NGS) IRD panel including 322 genes (Blueprint Genetics). Class 5 and 4 variants, according to ACMG guidelines, were considered pathogenic. Of the 279 samples tested, a confirmed genotype was determined in 182 (65%). The cohort was genetically heterogenous, including 65 different genes. The most prevailing were *ABCA4* (16.5%)*, RPGR* (6%), *CEP290* (6%), and *RS1* (5.5%). Other prevalent genes were *CACNA1F* (3%), *PROM1* (3%)*, CHM* (3%)*,* and *NYX* (3%). In 7% of the patients there was a discrepancy between the diagnosis made based on phenotypical or genotypical findings alone. To conclude, repeated DNA-analysis was beneficial also in previously tested patients and improved our ability to verify the genotype–phenotype association increasing the understanding of how visual impairment manifests, prognosis, and the inheritance pattern. Moreover, repeated testing using a widely available method could identify additional patients eligible for future gene-based therapies.

## 1. Introduction

Inherited retinal dystrophies (IRDs) are one of the most common causes of serious visual impairment in children and young adults in developed countries [1,2]. Until quite recently, IRDs have been untreatable, but during the last decades, extensive research concerning gene-based therapies [3,4,5,6,7] has evolved and the first gene augmentation therapy, Voretigene Neparvovec for treatment of *RPE65*-associated retinal dystrophies [8,9], was approved in USA in 2017 and in Europe 2018. Since the novel therapies such as gene augmentation/replacement, gene silencing, antisense oligonucleotides (AONs), and gene editing using the CRISPR/Cas9 system [3,5,6,7] are all based on correcting the specific genetic defect; verification of the genotype is essential nowadays. Moreover, there is a complicated overlap of genotypes and phenotypes in the sense that the same pathogenic genetic variant can cause several different clinical manifestations, e.g., either retinitis pigmentosa (RP), first engaging rods and, after some time, also cones, or Lebers congenital amaurosis (LCA) with early-onset rod and cone engagement, but also cone–rod dystrophy (CRD) with the cones affected primarily and rod secondarily [10,11,12]. Similarly, one phenotype, RP, can be caused by mutations in many different genes, (with over 60 currently known [13]). Concerning the whole spectrum of IRDs, over 300 causative genes [14] are known presently and they can be linked with over 50 separate phenotypes [13]. In this setting, careful mapping of the genetic cause of IRDs has become more important and lately, our ability to assess genotypes has improved significantly. Over the years, the procedure for DNA-analysis has evolved from single gene testing with the first gene associated with X-linked RP described in 1984 [15,16], via the APEX technique, to NGS panels and whole exome as well as whole genome sequencing (WES and WGS) [17]. Although modern procedures such as NGS panels, WES, and WGS are used, the diagnostic yield is not complete but ranges between 50–75% [14]. Thus, to optimize our ability to make the accurate diagnosis in each patient and thereby enable better understanding of the type of visual impairment, prognosis, and inheritance patterns, we must combine thorough clinical assessments and genetic testing. And, when it comes to finding patients eligible for gene-based therapies, genotyping is crucial, both the approved one and for therapies in clinical trials [3,4,6,7,18]. At the Department of Ophthalmology of Skåne University Hospital, we have, since the mid-1990s, had the ambition to verify the genotype in all patients, but that has not yet been fully possible. In this study, we wanted to investigate to what extent renewed comprehensive genetic testing with a widely available, broad NGS panel for IRDs, could verify the genotype in patients where previous genetic testing had been inconclusive and if confirmation of the genotype has an impact on the understanding of the clinical picture. Moreover, we aimed to describe the spectrum of genes encountered in a Swedish cohort of IRD patients.

## 2. Materials and Methods

### 2.1. Subjects

The study included 279 patients, with inconclusive previous DNA test results, from the retinitis pigmentosa research registry hosted at the Department of Ophthalmology, Skåne University Hospital, Lund, Sweden. Despite the name, the registry includes subjects with the whole spectrum of IRDs. The patients had made their first visit to the department between 1988 and 2015 and the initial appointment included a thorough clinical examination that mapped the phenotype carefully. Among the most prevalent diagnoses (based on the phenotype) were RP (94 subjects), CRD (38 subjects), Stargardt diseases (STGD) (24 subjects), X-linked juvenile retinoschisis (XLRS) (22 subjects), LCA (14 subjects), cone dystrophy (CD) (12 subjects), congenital stationary night blindness (CSNB) (11 subjects), macular dystrophy (11 subjects), and Usher syndrome (9 subjects). Previous DNA analyses were performed between 1995 and 2020 in cooperation with several collaborators, using both research laboratories and commercial facilities. Over time, the available techniques have developed from single gene tests and APEX panels to NGS panels and WES. Of the subjects, 122 had been tested with single-gene analysis, often including a range of genes on several occasions and in many different laboratories, while 157 of the patients that were investigated more recently had been tested with APEX—or NGS panels. A few cases with unsolved genotypes had also been tested with WES in addition to any of the other methods. In many cases, several DNA tests have been carried out over time. In this study, the term, inconclusive test results, means that either no pathogenic variant at all had been identified with previous tests or that only one pathogenic variant had been detected in a gene that is known to cause autosomal recessive disease. The study included 117 females and 162 males. They were all between 0 and 25 years of age at the time of their first visit (median 10 and mean 11 with standard deviation 6). Patients from widely distributed parts of Sweden are represented in the cohort, in which 60% had been referred from areas outside the department’s own region, Skåne. Hence, these results provide information about the genetic characteristics of Swedish IRD patients on a national level rather than on a regional level. The study was conducted in accordance with the Tenets of the Declaration of Helsinki and it was approved by the Ethical Committee for Medical Research at Lund University (nr 2015/602). All subjects gave their informed consent concerning the study including the DNA analysis.

### 2.2. Genetic Analysis

In 2021, DNA samples from all 279 patients were sent for renewed genetic testing with an NGS IRD panel including 322 genes at Blueprint Genetics, a College of American Pathologists- and Clinical Laboratory Improvement Amendments-certified laboratory. Investigated genes are listed in Table 1. Class 5 and 4 variants according to ACMG guidelines were considered pathogenic. In a few cases (Table 2), a class 3 variant was upgraded to a class 4 by the geneticists at Blueprint Genetics. The analysis also included assessment copy number variations (CNVs) as well as evaluation of the maternally inherited mitochondrial genome. In addition to the coding regions, the panel targeted 20 base pairs at the intron/exon boundaries and noncoding variants previously reported as disease-causing in association with IRD.

Bioinformatics and quality control were performed as follows. Base called raw sequencing data was transformed into FASTQ format using Illumina’s software (bcl2fastq) v2.20. Sequence reads of each sample were mapped to the human reference genome (GRCh37/hg19). Burrows–Wheeler Aligner (BWA-MEM) software was used for read alignment. Duplicate read marking, local realignment around indels, base quality v0.7.12 score recalibration and variant calling were performed using GATK algorithms (Sentieon) for nDNA. Variant data was annotated using a collection of tools (VcfAnno and VEP) with a variety of public variant databases including, but not limited to, gnomAD, ClinVar and HGMD. The median sequencing depth and coverage across the target regions for the tested sample were calculated based on MQ0 aligned reads. The sequencing run was included in process reference sample(s) for quality control, which passed our thresholds for sensitivity and specificity. The patient’s sample was subjected to thorough quality control measures including assessments for contamination and sample mix-up. Copy number variations (CNVs), defined as single exon or larger deletions or duplications (Del/Dups), were detected from the sequence analysis data using a commercially available bioinformatic pipeline CNVkit and a proprietary, in-house-developed deletion caller based on read depth to improve the detection of small CNVs. The difference between observed and expected sequencing depth at the targeted genomic regions was calculated and regions were divided into segments with variable DNA copy number. The expected sequencing depth was obtained by using other samples processed in the same sequence analysis as a guiding reference. The sequence data were adjusted to account for the effects of varying guanine and cytosine content.

### 2.3. DNA Extraction

DNA was extracted from venous blood drawn from the precubital vein. Buffy coats of nucleated cells obtained from anticoagulated blood (EDTA) were resuspended in 15 mL polypropylene centrifugation tubes with 3 mL of nuclei lysis buffer (10 mM Tris-HCl, 400 mM NaCl and 2 mM Na2EDTA, pH 8.2). The cell lysates were digested overnight at 37 °C with 0.2 mL of 10% SDS and 0.5 mL of a protease K solution (1 mg protease K in 1% SDS and 2 mM Na2EDTA). After digestion was complete, 1 mL of saturated NaCl (approximately 6 M) was added to each tube and shaken vigorously for 15 s, followed by centrifugation at 2500 rpm for 15 min. The precipitated protein pellet was left at the bottom of the tube and the supernatant containing the DNA was transferred to another 15 mL polypropylene tube. Exactly 2 volumes of room temperature absolute ethanol were added, and the tubes inverted numerous times until the DNA precipitated. The precipitated DNA strands were removed with a plastic spatula or pipette and transferred to a 1.5 mL microcentrifuge tube containing 100–200 pl TE buffer (10 mM Tris-HCl, 0.2 mM Na2EDTA, pH 7.5). The DNA was allowed to dissolve for 2 h at 37 °C before quantitating.

### 2.4. Ophthalmological Examinations

For assessment of overall retinal function, full-field electroretinograms (ffERG) according to the ISCEV standards at the time [19,20] were recorded in all of the patients. In subjects that had their appointment after the multifocal electroretinography (mfERG) technique had been introduced (from 2002), macular function was measured with mfERG according to the ISCEV standards of the time [21,22]. Best corrected visual acuity (BCVA) was tested monocularly on a decimal letter chart at 5 or 3 m (m) and visual fields were mapped with a Goldmann perimeter, likewise monocularly, with standardized objects V4e, I4e, 04e, 03e, and 02e. For structural analysis, fundus color and red free photographs, and during later years, optical coherence tomography (OCT) and autofluorescence (FAF) images were also obtained. Moreover, slit lamp and fundus examinations were conducted.

## 3. Results

Pathogenic class 4 or 5 genetic variants explaining the phenotype were found in 182 of the 279 (65%) samples that were re-analyzed with the NGS retinal dystrophy panel. A description of the pathogenic variants as well as data concerning age at first examination, gender, genotype, and phenotype at first examination are presented in Table 2. The cohort was genetically heterogenous showing disease -causing variants in 65 different genes (Figure 1 and Table 3). The most frequently mutated gene was the *ABCA4* gene with pathogen variants in 30 of the 182 (16.5%) cases with a verified genotype. Other prevalent causative genes in this Swedish cohort were *CEP290* (11 out of 182, 6%), *RPGR* (11 out of 182, 6%), *RS1* (10 out of 182, 5.5%), *CACNA1F* (6 out of 182, 3%), *CHM* (6 out of 182, 3%), *NYX* (6 out of 182, 3%), and *PROM1* (6 out of 182, 3%).

In 13 out of the 182 (7%) patients, there was a discrepancy between the diagnosis based on phenotypical or genotypical findings alone. The most common error was that CSNB initially was considered to be XLRS or choroideremia or that early choroideremia was mistaken for RP. In two cases, Bardet–Biedl syndrome initially was interpreted as achromatopsia before more general symptoms such as obesity and renal problems were apparent.

## 4. Discussion

Since gene-based treatments like gene augmentation/replacement [8,9,23,24,25,26,27,28,29,30,31], gene silencing, AONs [32,33], and gene editing using the CRISPR/Cas9 system [3,5,6,7] may be the future for patients with IRDs, confirmation of the genotype has become even more crucial during the last years. In our department, we have, since the 1990s, strived to both perform careful phenotyping and to verify the genotype in all our IRD patients. However, we have failed to identify the causative genetic background in quite a few of them and therefore, we wanted to investigate if it is beneficial to perform genetic re-testing with a widely available broad NGS panel for IRDs. WES or WGS could possibly have revealed more pathogenic variants, but in this study, we wanted to test a method that is affordable in a clinical setting and for the health care systems in different countries. In these patients, that had previously been investigated with various techniques such as single-gene analysis, APEX panels, NGS panels, and WES with inconclusive results, the renewed testing with a comprehensive NGS panel revealed the presence of the genotype in 182 individuals (65%). Thus, the success rate was approximately the same as the general yield described for first time-testing using NGS (50–71%) [34,35] although our subjects were selected unsolved cases. This means that it is of great value to re-test IRDs patients with unsolved genotypes using a broad NGS panel for IRDs. When it comes to the cases with compound heterozygosity it would, of course, be ideal to perform segregation analyses for all of them, but in this study, it was not possible to make contact with and test relatives of all of the patients. In many cases, NGS data could confirm that the variants were in trans and in all cases we were very careful in the interpretation of the genetic data only considering the genotype as causative if it was completely consistent with the phenotype. It is difficult to set a proper interval for DNA re-testing. In our study, the positive yield of testing for the patients with the shortest re-test interval (previously tested between 2016–2020 with APEX or NGS panels) was 32 out of 49 samples (65%), which means a similar positive success rate as for the whole group, indicating the usefulness of re-testing with quite short intervals.

When it comes to the prevalence of different causative genes in this Swedish cohort, which to our knowledge is the first larger cohort investigated concerning the genetic spectrum in Sweden, the *ABCA4* gene was the most common gene, encountered in 16.5% of the patients. This is in line with both an international estimate by Schneider et.al., 2022 called the Global Retinal Inherited Disease (GRID) dataset [13], and with reports from separate countries, although the absolute percentage varies slightly: GRID 25%, USA 14% [36], Canada 20% [36], Brazil 21% [37], Taiwan 15% [38], and Italy 26% [39]. Our second-most common genes were *RPGR* and *CEP290*, which were found in 6% of the patients, respectively. *RGPR* is also among the most prevalent genes in other studies; fourth-most common in the GRID dataset (3.4%) [13], in USA and Canada it was the third-most common gene (10% and 4%) [36], the fourth-most common in Brazil (5%) [37], fifth-most common in Taiwan (5%) [38], and in the Italian cohort, it was the third-most common gene (5%) [39]. *CEP 290*, on the other hand, is only represented to the same extent in the Brazilian cohort (5.5%) [37], while it is less common in the other cohorts (1–3%) [13,36,38,39]. Another difference is that *USH2A* is quite common in the other studies, being the second-most prevalent gene in the GRID dataset (15%) [13] as well as in the Italian (11%) [39] and the Canadian (6%) [36] cohorts, the third-most prevalent gene in Taiwan and Brazil (10% and 5%, respectively) [37,38], and found in 3% of American IRD patients [36], while it was found in only two of our 182 patients (1.1%) with a verified genotype. It is well known that genes have different prevalences in various countries and geographic areas, but most of the difference concerning the *USH2A* gene in our study can be explained by the fact that the patients with Usher syndrome type 2A are referred to us at an older age (mean age 39 at genotypic diagnosis in our registry) than the investigated group, since their visual decline becomes evident somewhat later in life. *EYS* was also among the more common genes in some of the other cohorts; e.g., second-most common among the Taiwanese subjects (12%) [38], third-most common in the GRID dataset (4.4%) [13,36], and was found in 4% of Brazilian IRD patients [37], while it was actually absent from our study as well as from the American and Canadian cohorts [36]. Concerning *RS1*, the setting was the opposite. It was among the more common genes in our cohort, verified in 5.5% of the subjects, but less prevalent in the other studies, in which it was described in only 0.5–2% of the patients [37,39,40] or was not specified at all [13,36]. Thus, these data indicate that to an extent, the same genes are the most prevalent across different cohorts with the exception of certain genes, e.g., *RS1, CEP290,* and *EYS*, that show more inconsistent distribution. This is of special interest when it comes to introducing gene-based therapies, since particular genes have a more urgent need to be dealt with in some populations than in others. Figure 2 shows the genotypic pattern of the 201 patients with established genotypes belonging to the same age group in the RP registry. In this group, 45 different genotypes were demonstrated. It can be noted from Figure 1 and Figure 2 that some genes such as *CNGB3, RHO, CLN3*, *BEST1, BCM*, and *GUCY2D* were quite well covered in the former analyses and not many new cases were encountered in the re-analysis. For *ABCA4, RPGR*, and *RS1*, new variants were discovered in rather many subjects although these genes were among the most prevalent causative genotypes also in the registry cohort and thus the coverage of those genes has improved. It is also noteworthy that *CEP290*, *CACNA1F*, *CHM*, and *NYX* are much better covered in the newer genetic work-up identifying many more subjects than in the registry cohort.

In the re-analyzed material, the gender distribution was slightly skewed, which can be explained by the occurrence of X-linked disorders that were encountered in 40 of the subjects (22% of the subjects with a verified genotype).

The basis for the choice of age range of 25 years or younger in the study was that younger patients are more suitable for future treatments, since many of the IRDs are progressive and thus, early detection is essential for enough viable retinal cells to be left for decent treatment results. Moreover, it is very important for young patients to obtain a correct diagnosis as early as possible, in order to enable adequate visual habilitation including visual aids, as well as fair expectations concerning the course of the visual impairment. In line with this, we can confirm the importance of a combined phenotypic and genotypic work-up, since in 7% of the patients, the result of genetic testing or clinical examinations alone led to different diagnoses, delaying correct counselling. For instance, in some early cases, X-linked congenital stationary night blindness (CSNB) due to *CACNA1F* variants was diagnosed as X-linked RP with the risk of giving the family incorrect information concerning the progression of the disease over time, since CSNB is a stationary and XLRP a progressive disorder. In some cases, the genetic result was also important for the confirmation of the inheritance pattern.

To conclude, renewed DNA-analysis was also beneficial in previously tested patients with inconclusive genetic test results, and it improved our ability to verify the genotype–phenotype association increasing the understanding of visual impairment, disease prognosis, and sometimes the inheritance pattern. Thus, repeated testing using a widely available method may identify additional patients eligible for future gene-based therapies.

## Figures and Tables

**Figure 1 genes-14-01413-f001:**
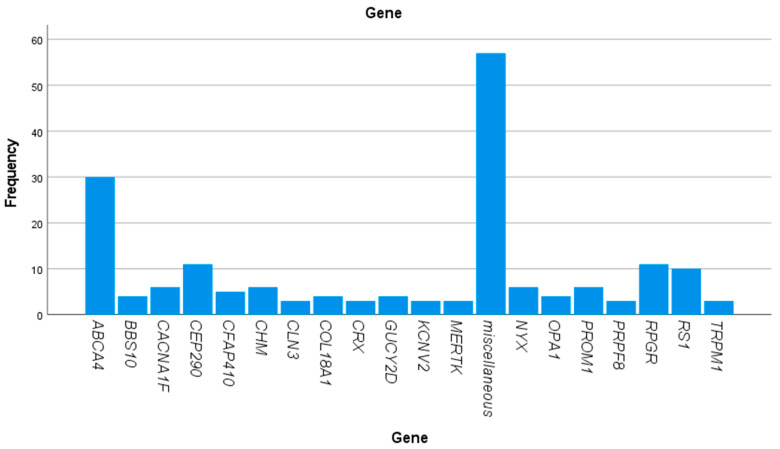
Showing the frequency of mutated genes that were found in the study.

**Figure 2 genes-14-01413-f002:**
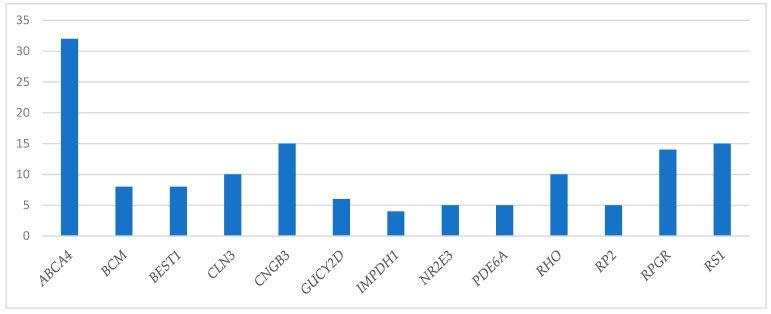
Showing the frequency of the most prevalent mutated genes in patients aged 0–25 years with established genotype in the retinitis pigmentosa research registry.

**Table 1 genes-14-01413-t001:** Listing the genes that were investigated in the NGS retinal dystrophy panel.

*ABCA4* *ABCC6* *ABHD12* *ACO2* *ADAM9* *ADAMTS18* *ADGRV1* *ADIPOR1* *AGBL5* *AHI1* *AIPL1* *ALMS1* *ARHGEF18* *ARL13B* *ARL2BP* *ARL3* *ARL6* *ARMC9* *ARSG* *ATF6* *ATOH7* *B9D1* *B9D2* *BBIP1* *BBS1* *BBS10* *BBS12* *BBS2* *BBS4* *BBS5* *BBS7* *BBS9* *BEST1* *C1QTNF5* *C21ORF2* *C2ORF71* *C5ORF42* *C8ORF37* *CA4* *CABP4*	*CACNA1F* *CACNA2D4* *CAPN5* *CC2D2A* *CDH23* *CDH3* *CDHR1* *CEP104* *CEP120* *CEP164* *CEP19* *CEP250* *CEP290* *CEP41* *CEP78* *CERK* *CHM* *CIB2* *CISD2* *CLN3* *CLRN1* *CNGA1* *CNGA3* *CNGB1* *CNGB3* *CNNM4* *COL11A1* *COL11A2* *COL18A1* *COL2A1* *COL9A1* *COL9A2* *COL9A3* *CPE* *CRB1* *CRX* *CSPP1* *CTC1* *CTNNA1* *CTNNB1*	*CWC27* *CYP4V2* *DFNB31* *DHDDS* *DHX38* *DRAM2* *DTHD1* *EFEMP1* *ELOVL4* *EMC1* *ESPN* *EYS* *FAM161A* *FDXR* *FLVCR1* *FRMD7* *FZD4* *GNAT1* *GNAT2* *GNB3* *GNPTG* *GPR179* *GRK1* *GRM6* *GUCA1A* *GUCY2D* *HAR* *HGSNAT* *HK1* *HMX* *IDH3A* *IDH3B* *IFT140* *IFT172* *IFT27* *IFT81* *IMPDH1* *IMPG1* *IMPG2* *INPP5E*	*INVS* *IQCB1* *JAG1* *KCNJ13* *KCNV2* *KIAA0556* *KIAA0586* *KIAA0753* *KIAA154* *KIF11* *KIF7* *KIZ* *KLHL7* *LCA5* *LRAT* *LRIT3* *LRP2* *LRP5* *LZTFL1* *MAK* *MERTK* *MFN2* *MFRP* *MFSD8* *MKKS* *MKS1* *MMACHC* *MT-ATP6* *MT-ATP8* *MT-CO1* *MT-CO2* *MT-CO3* *MT-CYB* *MT-ND1* *MT-ND2* *MT-ND3* *MT-ND4* *MT-ND4L* *MT-ND5* *MT-ND6*	*MT-RNR1* *MT-RNR2* *MT-TA* *MT-TC* *MT-TD* *MT-TE* *MT-TF* *MT-TG* *MT-TH* *MT-TI* *MT-TK* *MT-TL1* *MT-TL2* *MT-TM* *MT-TN* *MT-TP* *MT-TQ* *MT-TR* *MT-TS1* *MT-TS2* *MT-TT* *MT-TV* *MT-TW* *MT-TY* *MTTP* *MVK* *MYO7A* *NDP* *NEK2* *NMNAT1* *NPHP1* *NPHP3* *NPHP4* *NR2E3* *NR2F1* *NRL* *NYX* *OAT* *OFD1* *OPA1*	*OPA3* *OTX2* *P3H2* *PANK2* *PAX2* *PCDH15* *PCYT1A* *PDE6A* *PDE6B* *PDE6C* *PDE6D* *PDE6G* *PDE6H* *PDZD7* *PEX1* *PEX10* *PEX11B* *PEX12* *PEX13* *PEX14* *PEX16* *PEX19* *PEX2* *PEX26* *PEX3* *PEX5* *PEX6* *PEX7* *PHYH* *PISD* *PITPNM3* *PLA2G5* *PNPLA6* *POC1B* *POMGNT1* *PRCD* *PRDM13* *PROM1* *PRPF3* *PRPF31*	*PRPF4* *PRPF6* *PRPF8* *PRPH2* *PRPS1* *RAB28* *RAX2* *RBP3* *RBP4* *RCBTB1* *RD3* *RDH11* *RDH12* *RDH5* *REEP6* *RGR* *RGS9* *RGS9BP* *RHO* *RIMS1* *RLBP1* *ROM1* *RP1* *RP1L1* *RP2* *RPE65* *RPGR* *RPGRIP1* *RPGRIP1L* *RS1* *RTN4IP1* *SAG* *SAMD11* *SCAPER* *SCLT1* *SDCCAG8* *SEMA4A* *SLC24A1* *SLC25A46* *SLC7A14* *SNRNP200*	*SPATA7* *SPP2* *SRD5A3* *TCTN1* *TCTN2* *TCTN3* *TEAD1* *TIMM8A* *TIMP3* *TMEM107* *TMEM126A* *TMEM138* *TMEM216* *TMEM231* *TMEM237* *TMEM67* *TOPORS* *TRAF3IP1* *TREX1* *TRIM32* *TRPM1* *TSPAN12* *TTC21B* *TTC8* *TTLL5* *TTPA* *TUB* *TUBB4B* *TULP1* *USH1C* *USH1G* *USH2A* *VCAN* *VPS13B* *WDPCP* *WDR19* *WFS1* *YME1L1* *ZNF408* *ZNF423* *ZNF513*

**Table 2 genes-14-01413-t002:** Showing demographic data as well as genotype and phenotype for the patients with conclusive genetic re-testing.

Patient	Gender	Age at First Examination	Genotype	Description of Pathogenic Variants	Phenotype at Initial Examination
RP145LU	K	12	*ABC4A*	heterozygous for the missense variant, *ABCA4* c.2915C > A, p.(Thr972Asn), which is pathogenic and heterozygous for the frameshift variant, *ABCA4* c.4042del, p.(Thr1348Hisfs*41), which is likely pathogenic.	STGD
RP146LU	M	8	*ABC4A*	heterozygous for *ABCA4* c.2915C > A, p.(Thr972Asn), which is pathogenic and heterozygous for *ABCA4* c.4042del, p.(Thr1348Hisfs*41), which is likely pathogenic	STGD
RP173LU	K	14	*ABCA4*	heterozygous for *ABCA4* c.2894A > G, p.(Asn965Ser), which is pathogenic heterozygous for ABCA4 c.768G > T, p.(Val256=), which is pathogenic.	CD
RP125LU	M	4	*ABCA4*	heterozygous for *ABCA4* c.2588G > C, p.(Gly863Ala) classified as pathogenic and heterozygous for *ABCA4* c.5603A > T, p.(Asn1868Ile) classified as a risk factor	CRD
RP135LU	M	15	*ABCA4*	homozygous for a deletion, *ABCA4* c.2918 + 11_3522 + 86del which encompasses exons 20–23 classified as pathogenic	CRD
RP209LU	K	8	*ABCA4*	homozygous for *ABCA4* c.768G > T, p.(Val256=), which is pathogenic	CRD
RP253LU	K	10	*ABCA4*	homozygous for *ABCA4* c.319C > T, p.(Arg107*), which is pathogenic	CRD
RP5LU	K	7	*ABCA4*	homozygous for *ABCA4* c.768G > T, p.(Val256=) which is pathogenic	CRD
RP161LU	K	20	*ABCA4*	heterozygous for *ABCA4* c.4773 + 3A > G, which is pathogenic and heterozygous for ABCA4 c.768G > T, p.(Val256=), which is pathogenic	STGD
RP162LU	M	10	*ABCA4*	heterozygous for *ABCA4* c.6286G > A, p.(Glu2096Lys), which is pathogenic and heterozygous for *ABCA4* c.5461–10T > C, which is pathogenic and for *ABCA4* c.5603A > T, p.(Asn1868Ile), which is risk factor	STGD
RP170LU	K	14	*ABCA4*	heterozygous for *ABCA4* c.4139C > T, p.(Pro1380Leu), which is pathogenic. heterozygous for ABCA4 c.2894A > G, p.(Asn965Ser), which is pathogenic. heterozygous for ABCA4 c.5603A > T, p.(Asn1868Ile), which is a risk factor	STGD
RP171LU	K	15	*ABCA4*	heterozygous for *ABCA4* c.6181_6184del, p.(Thr2061Serfs*53), which is pathogenic, heterozygous for *ABCA4* c.3322C > T, p.(Arg1108Cys), which is pathogenic and heterozygous for ABCA4 c.5603A > T, p.(Asn1868Ile), which is a risk factor	STGD
RP18LU	K	12	*ABCA4*	heterozygous for *ABCA4* c.2894A > G, p.(Asn965Ser), which is pathogenic and heterozygous for *ABCA4* c.319C > T, p.(Arg107*), which is pathogenic	STGD
RP200LU	K	17	*ABCA4*	heterozygous for *ABCA4* c.3322C > T, p.(Arg1108Cys), which is pathogenic and heterozygous for *ABCA4* c.768G > T, p.(Val256=), which is pathogenic	STGD
RP206LU	K	18	*ABCA4*	heterozygous for *ABCA4* c.4601del, p.(Leu1534Trpfs*2), which is pathogenic, heterozygous for *ABCA4* c.2588G > C, p.(Gly863Ala), which is pathogenic and heterozygous for *ABCA4* c.5603A > T, p.(Asn1868Ile), which is risk factor	STGD
RP215LU	K	19	*ABCA4*	heterozygous for *ABCA4* c.5461-10T > C, which is pathogenic, heterozygous for *ABCA4* c.2894A > G, p.(Asn965Ser), which is pathogenic and heterozygous for *ABCA4* c.5603A > T, p.(Asn1868Ile), which is a risk factor	STGD
RP224LU	M	11	*ABCA4*	heterozygous for *ABCA4* c.4139C > T, p.(Pro1380Leu), which is pathogenic heterozygous for ABCA4 c.2599del, p.(Thr867Profs*34), which is likely pathogenic.	STGD
RP242LU	K	13	*ABCA4*	heterozygous for *ABCA4* c.2894A > G, p.(Asn965Ser), which is pathogenic and heterozygous for *ABCA4* c.1610G > A, p.(Arg537His), which is likely pathogenic	STGD
RP261LU	M	10	*ABCA4*	homozygous for *ABCA4* c.5584G > C, p.(Gly1862Arg), which is likely pathogenic	STGD
RP287LU	K	19	*ABCA4*	heterozygous for *ABCA4* c.6088C > T, p.(Arg2030*), which is pathogenic and heterozygous for *ABCA4* c.5882G > A, p.(Gly1961Glu), which is pathogenic	STGD
RP94LU	M	11	*ABCA4*	homozygous for *ABCA4* c.868C > T, p.(Arg290Trp), which is pathogenic	STGD
RP48LU	K	25	*ABCA4*	heterozygous for *ABCA4* c.3322C > T, p.(Arg1108Cys), which is pathogenic and heterozygous for *ABCA4* c.2894A > G, p.(Asn965Ser), which is pathogenic	STGD
RP85LU	K	25	*ABCA4*	homozygous for *ABCA4* c.5882G > A, p.(Gly1961Glu), which is pathogenic	STGD
RP191LU	K	16	*ABCA4*	heterozygous for *ABCA4* c.1804C > T, p.(Arg602Trp), which is pathogenic, heterozygous for *ABCA4* c.3113C > T, p.(Ala1038Val), which is pathogenic, heterozygous for *ABCA4* c.1622T > C, p.(Leu541Pro), which is pathogenic and heterozygous for *ABCA4* c.5603A > T, p.(Asn1868Ile), which is a risk factor	CD
RP21LU	K	18	*ABCA4*	homozygous ABCA4c.5882G > A, p.(Gly1961Glu pathogenic. homozygous for ABCA4 c.634C > T, p.(Arg212Cys), which is pathogenic	CD
RP22LU	K	10	*ABCA4*	heterozygous for *ABCA4* c.4773 + 1G > A, which is pathogenic and heterozygous for *ABCA4* c.53G > A, p.(Arg18Gln), which is pathogenic	CRD
RP172LU	K	8	*ABCA4*	homozygous for *ABCA4* c.3113C > T, p.(Ala1038Val), which is pathogenic and homozygous for *ABCA4* c.1622T > C, p.(Leu541Pro), which is pathogenic	CRD
RP20LU	K	14	*ABCA4*	heterozygous for *ABCA4* c.5413A > G, p.(Asn1805Asp), which is pathogenic and heterozygous for *ABCA4* c.6159G > A, p.(Trp2053*), which is likely pathogenic	STGD
RP34LU	M	18	*ABCA4*	heterozygous for *ABCA4* c.5461-10T > C, which is pathogenic, heterozygous for *ABCA4* c.5196 + 1137G > A, which is pathogenic and heterozygous for *ABCA4* c.5603A > T, p.(Asn1868Ile), which is risk factor	STGD
RP41LU	M	15	*ABCA4*	heterozygous for *ABCA4* c.6079C > T, p.(Leu2027Phe), which is pathogenic and heterozygous for *ABCA4* c.4139C > T, p.(Pro1380Leu), which is pathogenic	STGD
RP273LU	M	1	*AIPL1*	heterozygous for *AIPL1* c.834G > A, p.(Trp278*), which is pathogenic and heterozygous for *AIPL1* c.537del, p.(Val180Serfs*29), which is likely pathogenic	LCA
RP181LU	K	18	*BBS1*	homozygous for *BBS1* c.1169T > G, p.(Met390Arg), which is pathogenic	RP
RP12LU	M	3	*BBS10*	homozygous *BBS10* c.271dup, p.(Cys91Leufs*5), which is pathogenic	Bardet–Biedl
RP154LU	K	13	*BBS10*	homozygous for *BBS10* c.1244del, p.(His415Leufs*16), which is pathogenic	Bardet–Biedl
RP155LU	K	14	*BBS10*	homozygous for *BBS10* c.271dup, p.(Cys91Leufs*5), which is pathogenic	Bardet–Biedl
RP190LU	K	8	*BBS10*	homozygous for *BBS10* c.271dup, p.(Cys91Leufs*5), which is pathogenic	Bardet–Biedl
RP236LU	M	12	*BBS5*	homozygous for *BBS5* c.790G > A, p.(Gly264Arg), which is pathogenic	achromatopsia
RP30LU	K	8	*BBS5*	homozygous for *BBS5* c.790G > A, p.(Gly264Arg), which is pathogenic	achromatopsia
RP76LU	M	16	*BBS9*	homozygous for *BBS9* c.1561C > T, p.(Arg521*), which is pathogenic	Bardet–Biedl
RP1LU	M	10	*CACNA1F*	hemizygous for *CACNA1F* c.4156C > T, p.(Gln1386*) which is likely pathogenic	CHM
RP205LU	M	6	*CACNA1F*	hemizygous for *CACNA1F* c.3895C > T, p.(Arg1299*), which is pathogenic	CSNB
RP195LU	M	7	*CACNA1F*	hemizygous for *CACNA1F* c.4134–1G > C, which is pathogenic	XLRS
RP23LU	M	8	*CACNA1F*	hemizygous for *CACNA1F* c.3542_3548del, p.(Tyr1181Cysfs*5), which is likely pathogenic	XLRS
RP166LU	M	8	*CACNA1F*	hemizygous for *CACNA1F* c.952_954del, p.(Phe318del), which is pathogenic	XLRS
RP50LU	M	2	*CACNA1F*	hemizygous for *CACNA1F* c.2071C > T, p.(Arg691*), which is pathogenic	XLRS
RP65LU	M	19	*CACNA2D4*	homozygous for *CACNA2D4* c.1564C > T, p.(Arg522*), which is likely pathogenic	CD
RP123LU	K	6	*CDH23*	homozygous for *CDH2* c.8733del, p.(Asp2911Glufs*41), which is likely pathogenic	Usher
RP108LU	M	12	*CDH3*	heterozygous for *CDH3* c.1795 + 1G > A, which is likely pathogenic and heterozygous for *CDH3* c.1643C > G, p.(Pro548Arg), which is a VUS; however, this variant is absent in control populations and predicted to be deleterious via in silico tools and NGS data strongly suggest that these variants are in trans, thus interpreted as causative	macular dystrophy
RP221LU	K	19	*CDHR1*	heterozygous for *CDHR1* c.783G > A, p.(Pro261=), which is pathogenic and heterozygous for *CDHR1* c.2522_2528del, p.(Ile841Serfs*119), which is pathogenic	RP
RP106LU	M	1	*CEP290*	heterozygous for *CEP290* c.4661_4663del, p.(Glu1554del), which is pathogenic and heterozygous for *CEP290* c.2052 + 1_2052 + 2del, which is pathogenic	LCA
RP116LU	K	3	*CEP290*	heterozygous for *CEP290* c.4661_4663del, p.(Glu1554del), pathogenic. heterozygous for CEP290 c.955del, p.(Ser319Leufs*16), likely pathogenic	LCA
RP137LU	K	1	*CEP290*	heterozygous for *CEP290* c.2991 + 1655A > G, which is pathogenic and heterozygous for *CEP290* c.1992del, p.(Pro665Leufs*10), which is pathogenic	LCA
RP150LU	M	1	*CEP290*	homozygous for *CEP290* c.2991 + 1655A > G, which is pathogenic	LCA
RP156LU	K	1	*CEP290*	heterozygous for *CEP290* c.2991 + 1655A > G, which is pathogenic and heterozygous for *CEP290* c.384_387del, p.(Asp128Glufs*34), which is pathogenic	LCA
RP157LU	K	0	*CEP290*	heterozygous for *CEP290* c.2991 + 1655A > G, which is pathogenic. heterozygous for CEP290 c.170C > A, p.(Ser57*), which is likely pathogenic	LCA
RP249LU	K	7	*CEP290*	heterozygous for *CEP290* c.3249dup, p.(Arg1084Thrfs*11), which is pathogenic and heterozygous for *CEP290* c.1065 + 1G > A, which is likely pathogenic	LCA
RP294LU	M	19	*CEP290*	heterozygous for *CEP290* c.2991 + 1655A > G, which is pathogenic and heterozygous for *CEP290* c.384_387del, p.(Asp128Glufs*34), which is pathogenic	LCA
RP66LU	K	0	*CEP290*	heterozygous for *CEP290* c.2991 + 1655A > G, which is pathogenic and heterozygous for CEP290 c.1681C > T, p.(Gln561*), which is likely pathogenic.	LCA
RP82LU	M	1	*CEP290*	heterozygous for *CEP290* c.2991 + 1655A > G, which is pathogenic and heterozygous for *CEP290* c.1992del, p.(Pro665Leufs*10), which is pathogenic	LCA
RP262LU	M	5	*CEP290*	heterozygous for *CEP290* c.4438–3del, which is pathogenic and heterozygous for CEP290 c.164_167del, p.(Thr55Serfs*3), which is pathogenic	RP
RP258LU	K	5	*CFAP410*	homozygous for *CFAP410* c.33_34insAGCTGCACAGCGTGCA, p.(Ala12Serfs*60), which is pathogenic	CD
RP59LU	K	5	*CFAP410*	homozygous for *CFAP410* c.218G > C, p.(Arg73Pro) which is pathogenic	CD
RP64LU	K	11	*CFAP410*	homozygous for deletion *CFAP410* c.(?_-1)_(*1_?)del, which is pathogenic	CD
RP80LU	K	14	*CFAP410*	homozygous for a deletion *CFAP410* c.(?_-1)_(*1_?)del, which encompasses the whole *CFAP410* gene and is pathogenic	CD
RP256LU	K	10	*CFAP410*	homozygous for *CFAP410* c.218G > C, p.(Arg73Pro), which is pathogenic	RP
RP288LU	M	24	*CHM*	hemizygous for *CHM* c.1244 + 1G > C, which is likely pathogenic	CHM
RP49LU	M	16	*CHM*	hemizygous for *CHM* c.1144G > T, p.(Glu382*), which is pathogenic	CHM
RP257LU	M	18	*CHM*	hemizygous for a deletion *CHM* c.(314 + 1_315 − 1)_(1166 + 1_1167 − 1)del, which encompasses exons 5–8 of *CHM,* classified as pathogenic	RP
RP264LU	M	10	*CHM*	hemizygous for a 6 Mb deletion, seq[GRCh37] del(X)(q21.1q21.2), chrX:g.79270061–85302755del, encompassing the entire panel gene *CHM* and classified as pathogenic.	RP
RP129LU	K	8	*CHM*	heterozygous for *CHM* c.1411del, p.(Gln471Argfs*5), which is likely pathogenic	CHM carrier
RP42LU	K	13	*CHM*	heterozygous for *CHM* c.1144G > T, p.(Glu382*), which is pathogenic	CHM carrier
RP227LU	K	7	*CLN3*	homozygous for deletion *CLN3* c.(460 + 1_461 − 1)_(677 + 1_678 − 1)del, which encompasses exons 8–9 of *CLN3* and is classified as pathogenic	CLN3
RP234LU	M	7	*CLN3*	homozygous for deletion *CLN3* c.(460 + 1_461 − 1)_(677 + 1_678 − 1)del, which encompasses exons 8–9 of *CLN3* and is classified as pathogenic	CLN3
RP220LU	M	6	*CLN3*	homozygous for a deletion *CLN3* c.(460 + 1_461 − 1)_(677 + 1_678 − 1)del, which encompasses exons 8–9 of *CLN3* and is classified as pathogenic	RP
RP68LU	M	18	*CNGB1*	heterozygous for *CNGB1* c.2957A > T, p.(Asn986Ile), which is pathogenic and heterozygous for *CNGB1* c.2293C > T, p.(Arg765Cys), which is likely pathogenic	RP
RP26LU	M	10	*CNGB3*	heterozygous for *CNGB3* c.1285dup, p.(Ser429Phefs*33), which is pathogenic and heterozygous for *CNGB3* c.819_826del, p.(Arg274Valfs*13), which is pathogenic	achromatopsia
RP27LU	M	1	*CNGB3*	heterozygous for *CNGB3* c.1148del, p.(Thr383Ilefs*13), which is pathogenic and heterozygous for *CNGB3* c.1643G > T, p.(Gly548Val), which is VUS, however, *CNGB3* c.1643G > T, p.(Gly548Val) is absent in control populations and predicted to be deleterious by insilico tools and thus compound heterozygosity of the variants would explain the phenotype	achromatopsia
RP15LU	K	6	*COL18A1*	homozygous for *COL18A1* c.2157 + 2T > C, which is likely pathogenic	Knobloch syndrome
RP132LU	M	2	*COL18A1*	homozygous for *COL18A1* c.3514_3515del, p.(Leu1172Valfs*72), which is pathogenic	LCA
RP244LU	K	12	*COL18A1*	homozygous for *COL18A1* c.874del, p.(Glu292Lysfs*17), which is likely pathogenic	macular dystrophy
RP207LU	M	10	*COL18A1*	heterozygous for *COL18A1* c.3666_3682del, p.(Ala1223Glnfs*19), which is likely pathogenic and heterozygous for *COL18A1* c.3809 + 2T > C, which is likely pathogenic	vitreoretinal dystrophy
RP140LU	M	13	*CRX*	heterozygous for *CRX* c.413del, p.(Ile138Thrfs*49), which is pathogenic	CRD
RP92LU	K	5	*CRX*	heterozygous for CRX c.413del, p.(Ile138Thrfs*49), which is pathogenic	CRD
RP117LU	M	4	*CRX*	heterozygous frameshift variant *CRX* c.413del, p.(Ile138Thrfs*49) which is pathogenic	RP
RP114LU	K	1	*GUCA1A*	heterozygous for *GUCA1A* c.332A > T, p.(Glu111Val), which is likely pathogenic	CRD
RP134LU	M	13	*GUCY2D*	heterozygous for *GUCY2D* c.2377del, p.(Glu793Asnfs*42), which is likely pathogenic and heterozygous for GUCY2D c.1567-17T > A, which is a VUS, however, these *GUCY2D* variants are consistent with the patient’s phenotype, and *GUCY2D* c.1567-17T > A is rare in control populations and predicted to affect splicing by in silico tools, thus compound heterozygosity of the variants could explain the phenotype	CRD
RP148LU	M	3	*GUCY2D*	heterozygous for *GUCY2D* c.2944 + 1del, which is pathogenic and heterozygous for GUCY2D c.2965G > C, p.(Val989Leu), which is a VUS, however, these *GUCY2D* variants are consistent with the patient’s phenotype, and *GUCY2D* c.2965G > C, p.(Val989Leu) is absent in control populations and predicted to be deleterious by in silico tools, NGS data suggests that these variants are in trans in thispatient, which could explain the patient’s clinical presentation	CRD
RP176LU	K	18	*GUCY2D*	heterozygous for *GUCY2D* c.2944 + 1del, which is pathogenic. heterozygous for GUCY2D c.1982G > T, p.(Gly661Val), which is a VUS, however, these *GUCY2D* variants are consistent with the patient’s phenotype, and GUCY2D c.1982G > T, p.(Gly661Val) is absent in control populations and predicted to be deleterious by in silico tools, compound heterozygosity of the variants would explain the patient’s clinical presentation	CRD
RP24LU	M	6	*GUCY2D*	heterozygous for *GUCY2D* c.2302C > T, p.(Arg768Trp), which is pathogenic and heterozygous for *GUCY2D* c.1567-17T > A, which is a VUS, however, these *GUCY2D* variants are consistent with the patient’s phenotype, and *GUCY2D* c.1567-17T > A is rare in control populations and predicted to affect splicing by in silico tools, compound heterozygosity of the variants could explain the patient’s clinical presentation	CRD
RP217LU	M	16	*IMPDH1*	heterozygous for *IMPDH1* c.931G > A, p.(Asp311Asn), which is pathogenic	RP
RP40LU	K	2	*IQCB1*	heterozygous for *IQCB1* c.1332G > A, p.(Trp444*), which is pathogenic and heterozygous for *IQCB1* c.424_425del, p.(Phe142Profs*5), which is pathogenic	Senior–Loken
RP104LU	M	16	*KCNV2*	homozygous for the nonsense variant *KCNV2* c.427G > T, p.(Glu143*), which is pathogenic	CRD
RP112LU	K	13	*KCNV2*	homozygous for *KCNV2* c.427G > T, p.(Glu143*), which is pathogenic	CRD
RP98LU	M	11	*KCNV2*	homozygous for *KCNV2* c.427G > T, p.(Glu143*), which is pathogenic	CRD
RP43LU	K	1	*KIF11*	heterozygous for *KIF11* c.1985T > A, p.(Leu662*), which is pathogenic	microcephaly and RD
RP78LU	K	20	*KLHL7*	heterozygous for *KLHL7* c.422T > C, p.(Val141Ala) which is likely pathogenic	RP
RP91LU	K	21	*LRAT*	homozygous for *LRAT* c.470T > C, p.(Leu157Pro), which is likely pathogenic	EORD
RP266LU	M	17	*LRAT*	homozygous for *LRAT* c.470T > C, p.(Leu157Pro), which is likely pathogenic	RP
RP121LU	M	9	*MERTK*	homozygous for *MERTK* c.2302G > A, p.(Ala768Thr), which is pathogenic	RP
RP290LU	M	13	*MERTK*	homozygous for *MERTK* c.1960 + 1G > A, which is likely pathogenic	RP
RP71LU	M	12	*MERTK*	heterozygous for *MERTK* c.345C > G, p.(Cys115Trp), which is pathogenic and heterozygous for *MERTK* c.1377_1379delinsAGCC, p.(Arg460Alafs*15), which is likely pathogenic	RP
RP238LU	M	19	*MFN2*	heterozygous for deletion *MFN2* c.(474 + 1_475 − 1)_(816 + 1_817 − 1)del, which encompasses exons 6–8 of *MFN2*. This alteration is classified as likely pathogenic	macular dystrophy
RP4LU	K	17	*MFRP*	homozygous for *MFRP* c.1090_1091del, p.(Thr364Glnfs*27), which is pathogenic	RP
RP203LU	M	4	*MYO7A*	heterozygous for *MYO7A* c.1556G > A, p.(Gly519Asp), which is pathogenic and heterozygous for *MYO7A* c.3719G > A, p.(Arg1240Gln), which is pathogenic	Usher
RP300LU	M	2	*MYO7A*	heterozygous for *MYO7A* c.401T > A, p.(Ile134Asn), which is pathogenic. heterozygous for MYO7A c.6558 + 1G > T, which is likely pathogenic	Usher
RP115LU	K	1	*NMNAT1*	heterozygous for *NMNAT1* c.196C > T, p.(Arg66Trp) and heterozygous for *NMNAT1* c.769G > A, p.(Glu257Lys), which are both pathogenic	LCA
RP90LU	K	9	*NPHP1*	homozygous for a deletion *NPHP1* c.(?_-1)_(*1_?)del, which encompasses the whole *NPHP1* gene, which is classified as pathogenic	RP and renal failure
RP194LU	K	11	*NR2E3*	heterozygous for *NR2E3* c.119-2A > C and heterozygous for NR2E3 c.349 + 5G > C, which are both pathogenic	RP
RP216LU	M	5	*NR2E3*	heterozygous for *NR2E3* c.119-2A > C, and *NR2E3* c.932G > A, p.(Arg311Gln), which are both pathogenic	RP
RP136LU	M	6	*NYX*	hemizygous for *NYX* c.85_108del, p.(Arg29_Ala36del), which is pathogenic	CSNB
RP138LU	M	9	*NYX*	hemizygous for *NYX* c.559_560delinsAA, p.(Ala187Lys), which is likely pathogenic	CSNB
RP84LU	M	2	*NYX*	hemizygous for *NYX* c.559_560delinsAA, p.(Ala187Lys), which is likely pathogenic	CSNB
RP185LU	M	8	*NYX*	hemizygous for *NYX* c.559_560delinsAA, p.(Ala187Lys), which is likely pathogenic	CSNB
RP201LU	M	6	*NYX*	hemizygous for *NYX* c.559_560delinsAA, p.(Ala187Lys), which is likely pathogenic	CSNB
RP233LU	M	5	*NYX*	hemizygous for *NYX* c.559_560delinsAA, p.(Ala187Lys), which is likely pathogenic	CSNB
RP160LU	M	4	*OPA1*	heterozygous for *OPA1* c.983A > G, p.(Lys328Arg), which is pathogenic	optic atrophy
RP184LU	M	8	*OPA1*	heterozygous for a deletion *OPA1* c.(?_-1)_(*1_?)del, which encompasses the whole *OPA1* gene	optic atrophy
RP286LU	K	24	*OPA1*	heterozygous for *OPA1* c.2497-4_2557del, which is likely pathogenic	optic atrophy
RP107LU	M	2	*OPA1*	heterozygous for *OPA1* c.703C > T, p.(Arg235*), which is pathogenic	RP
RP149LU	M	13	*OTX2*	heterozygous for *OTX2* c.483dup, p.(Asp162Argfs*25), which is likely pathogenic	EORD
RP131LU	K	9	*PANK2*	heterozygous for *PANK2* c.981 + 1G > C, which is likely pathogenic and heterozygous for *PANK2* c.1512dup, p.(Ala505Serfs*7), which is likely pathogenic	RP and neurological symptoms
RP44LU	M	9	*PCARE*	homozygous for *PCARE* c.1541del, p.(Pro514Hisfs*27), which is pathogenic	RP
RP83LU	K	18	*PCDH15*	heterozygous for *PCDH15* c.310del, p.(Asp104Ilefs*6) and *PCDH15* c.3761dup, p.(Asn1254Lysfs*54), which are likely pathogenic	Usher
RP99LU	M	2	*PCDH15*	homozygous for *PCDH15* c.3441dup, p.(Phe1148Ilefs*8), which is pathogenic	Usher
RP120LU	M	12	*PDE6B*	heterozygous for *PDE6B* c.1580T > C, p.(Leu527Pro) and *PDE6B* c.2193 + 1G > A which are both pathogenic	RP
RP235LU	M	2	*PDE6C*	heterozygous for *PDE6C* c.826C > T, p.(Arg276*) and *PDE6C* c.2457T > A, p.(Tyr819*), which are both likely pathogenic	CD
RP2LU	M	8	*PNPLA6*	heterozygous for *PNPLA6* c.(2143 + 1_2144-1)_(2351 + 1_2352 − 1)del, which is likely pathogenic and heterozygous for *PNPLA6* c.3625T > C, p.(Trp1209Arg), which is a VUS; however, these *PNPLA6* variants are consistent with the patient’s phenotype, and *PNPLA6* c.3625T > C, p.(Trp1209Arg) is rare in control populations and predicted to be deleterious by in silico tools, compound heterozygosity of the variants would explain the patient’s clinical presentation	RP
RP37LU	K	12	*POC1B*	heterozygous for *POC1B* c.1331_1332dup, p.(Thr445Argfs*10), which is pathogenic and heterozygous for *POC1B* c.52A > T, p.(Lys18*), which is likely pathogenic	achromatopsia
RP177LU	M	4	*POMGNT1*	homozygous for *POMGNT1* c.1539 + 1G > A, which is pathogenic	Muscle–Eye–Brain Disease
RP67LU	K	10	*PROM1*	heterozygous for a deletion *PROM1* c.(?-1)_(220 + 1_221 − 1)del, which encompasses exon 1 of *PROM1* and is classified as likely pathogenic	CD
RP198LU	M	12	*PROM1*	heterozygous for *PROM1* c.2050C > T, p.(Arg684*), which is pathogenic and heterozygous for *PROM1* c.1632G > T, p.(Gly544=), which is pathogenic	CRD
RP29LU	M	18	*PROM1*	heterozygous for a deletion *PROM1* c.(?-1)_(220 + 1_221 − 1)del, which encompasses exon 1 of *PROM1* and is classified as likely pathogenic	RP
RP32LU	M	23	*PROM1*	heterozygous for a deletion *PROM1* c.(?-1)_(220 + 1_221 − 1)del, which encompasses exon 1 of *PROM1* and is likely pathogenic	RP
RP31LU	K	10	*PROM1*	homozygous for *PROM1* c.1909C > T, p.(Gln637*), which is likely pathogenic	RP
RP47LU	K	18	*PROM1*	homozygous for *PROM1* c.1909C > T, p.(Gln637*), which is likely pathogenic	RP
RP245LU	M	15	*PRPF31*	heterozygous for a deletion *PRPF31* c.(?_-396)_(*1_?)del, which encompasses the whole *PRPF31* gene and is classified as pathogenic	RP
RP270LU	K	22	*PRPF31*	heterozygous for a deletion *PRPF31* c.(?_-396)_(*1_?)del, which encompasses the entire *PRPF31* gene and is classified as pathogenic	RP
RP103LU	M	12	*PRPF8*	heterozygous for *PRPF8* c.5804G > A, p.(Arg1935His), which is pathogenic	RP
RP188LU	K	20	*PRPF8*	heterozygous for *PRPF8* c.6901C > T, p.(Pro2301Ser), which is pathogenic	RP
RP248LU	M	5	*PRPF8*	heterozygous for *PRPF8* c.6926A > T, p.(His2309Leu), which is likely pathogenic	RP
RP179LU	M	14	*PRPH2*	heterozygous for *PRPH2* c.633C > G, p.(Phe211Leu), which is pathogenic	RP
RP8LU	M	19	*RDH12*	homozygous for *RDH12* c.481C > T, p.(Arg161Trp), which pathogenic	CRD
RP7LU	K	9	*RDH5*	homozygous for *RDH5* c.382G > A, p.(Asp128Asn), which is pathogenic	Fundus albipunctatus
RP11LU	M	10	*RHO*	heterozygous for *RHO* c.541G > A, p.(Glu181Lys), which is pathogenic	Aaland eye disease
RP74LU	K	19	*RLBP1*	Homozygous for *RLBP1* c.286_297del p.(Phe96_Phe99del), which is pathogenic	RP with maculopathy
RP292LU	K	8	*RP1*	heterozygous for *RP1* c.1498_1499del, p.(Met500Valfs*7), which is pathogenic and heterozygous for *RP1* c.1601_1604del, p.(Lys534Argfs*11), which is likely pathogenic	RP
RP189LU	M	13	*RP1*	heterozygous for *RP1* c.271del, p.(Ser91Alafs*25), and *RP1* c.753C > A, p.(Tyr251*), which are both likely pathogenic	RP
RP124LU	M	13	*RP1L1*	heterozygous for *RP1L1* c.133C > T, p.(Arg45Trp), which is pathogenic	macular dystrophy
RP219LU	M	16	*RP2*	hemizygous for *RP2* c.400C > T, p.(Gln134*), which is pathogenic	RP
RP86LU	K	1	*RPE65*	heterozygous for *RPE65* c.886dup, p.(Arg296Lysfs*7), which is pathogenic and heterozygous for *RPE65* c.612C > A, p.(Tyr204*), which is likely pathogenic	RP
RP16LU	K	16	*RPGR*	heterozygous for *RPGR* c.2641G > T, p.(Glu881*), which is likely pathogenic	RP
RP225LU	M	15	*RPGR*	hemizygous for *RPGR* c.764C > T, p.(Thr255Ile), which is likely pathogenic	RP
RP60LU	M	20	*RPGR*	hemizygous for *RPGR* c.2730_2731del, p.(Glu911Glyfs*167), which is pathogenic	RP
RP63LU	M	16	*RPGR*	hemizygous for *RPGR* c.2405_2406del, p.(Glu802Glyfs*32), which is pathogenic	RP
RP113LU	M	8	*RPGR*	hemizygous for *RPGR* c.2252_2255del, p.(Lys751Argfs*63), which is pathogenic	RP
RP192LU	M	10	*RPGR*	hemizygous for the deletion *RPGR* c.(1572 + 1_1573 − 1)_(*1_?)del, which encompasses exons 14–19 of *RPGR* and is classified as pathogenic	RP
RP199LU	M	12	*RPGR*	hemizygous for *RPGR* c.1573-8A > G, which is likely pathogenic	RP
RP222LU	M	15	*RPGR*	hemizygous for *RPGR* c.2426_2427del, p.(Glu809Glyfs*25), which is pathogenic	RP
RP251LU	M	20	*RPGR*	hemizygous for a deletion *RPGR* c.(1414 + 1_1415 − 1)_(*1_?)del, which encompasses exons 12–15 of *RPGR* and is classified as pathogenic	RP
RP100LU	M	8	*RPGR*	hemizygous for a deletion *RPGR* c.(778 + 1_779 − 1)_(1245 + 1_1246 − 1)del, which encompasses exons 8–10 of *RPGR* and is classified as pathogenic.	RP
RP52LU	M	16	*RPGR*	hemizygous for *RPGR* c.3300_3301del, p.(His1100Glnfs*10), which is pathogenic	macular dystrophy
RP167LU	M	9	*RS1*	hemizygous for *RS1* c.416del, p.(Gln139Argfs*10) which is classified as pathogenic	XLRS
RP168LU	M	5	*RS1*	hemizygous for a deletion *RS1* c.(?_-1)_(52 + 1_53 − 1)del, encompassing exon 1 of *RS1*, which is classified as pathogenic	XLRS
RP193LU	M	18	*RS1*	hemizygous for *RS1* c.214G > A, p.(Glu72Lys), which is pathogenic	XLRS
RP223LU	M	10	*RS1*	hemizygous for deletion *RS1* c.(?_-1)_(52 + 1_53 − 1)del, which encompasses exon 1 of *RS1* and is pathogenic	XLRS
RP226LU	M	19	*RS1*	hemizygous for a deletion *RS1* c.(?_-1)_(52 + 1_53 − 1)del, encompassing exon 1 of *RS1,* classified as pathogenic	XLRS
RP33LU	M	10	*RS1*	hemizygous for *RS1* c.149G > A, p.(Trp50*), which is pathogenic	XLRS
RP35LU	M	10	*RS1*	hemizygous for RS1 c.366G > A, p.(Trp122*), which is pathogenic	XLRS
RP36LU	M	11	*RS1*	hemizygous for a deletion *RS1* c.(?_-1)_(52 + 1_53 − 1)del, encompassing exon 1 of *RS1*, which is pathogenic	XLRS
RP79LU	M	19	*RS1*	hemizygous for a deletion *RS1* c.(?_-1)_(52 + 1_53 − 1)del, encompassing exon 1 of *RS1*, which pathogenic	XLRS
RP95LU	M	6	*RS1*	hemizygous for a deletion *RS1* c.(?_-1)_(52 + 1_53 − 1)del, encompassing exon 1 of *RS1,* which is pathogenic	XLRS
RP232LU	K	15	*TIMM8A*	heterozygous for *TIMM8A* c.116del, p.(Met39Argfs*26), which is pathogenic	carrier of Mohr–Tranebjaerg syndrome
RP126LU	K	1	*TRPM1*	heterozygous for the deletion *TRPM1* c.(−64 + 1_−63 − 1)_(899 + 1_900 − 1)del, encompassing exons 2 (first coding exon) to 7, which is classified as pathogenic and heterozygous for *TRPM1* c.3607_3608del, p.(Glu1203Asnfs*11), which is likely pathogenic	CSNB
RP13LU	K	7	*TRPM1*	homozygous for *TRPM1* c.2629C > T, p.(Arg877*), which is pathogenic	CSNB
RP243LU	M	8	*TRPM1*	homozygous for *TRPM1* c.2629C > T, p.(Arg877*), which is pathogenic	CSNB
RP169LU	M	3	*TULP1*	homozygous for *TULP1* c.148del, p.(Glu50Asnfs*59), which is pathogenic	LCA
RP75LU	K	11	*TULP1*	homozygous for *TULP1* c.1153G > A, p.(Gly385Arg), which is pathogenic	RP
RP61LU	K	7	*USH1C*	heterozygous for *USH1C* c.496 + 1G > T, and *USH1C* c.238dup, p.(Arg80Profs*69), which are pathogenic	Usher
RP197LU	M	8	*USH2A*	heterozygous for *USH2A* c.10450C > T, p.(Arg3484*), and *USH2A* c.779T > G, p.(Leu260*), whichare pathogenic	Usher
RP291LU	M	1	*USH2A*	heterozygous for *USH2A* c.8682-9A > G, which is pathogenic and heterozygous for *USH2A* c.1070_1071del, p.(Asn357Serfs*9), which is likely pathogenic	Usher
RP159LU	K	13	*WFS1*	heterozygous for *WFS1* c.1673G > A, p.(Arg558His), which is pathogenic and heterozygous for *WFS1* c.2149G > A, p.(Glu717Lys), which is pathogenic	optic atrophy
RP9LU	K	17	*WFS1*	heterozygous for *WFS1* c.1673G > A, p.(Arg558His), and *WFS1* c.2149G > A, p.(Glu717Lys), which are pathogenic	optic atrophy

**Table 3 genes-14-01413-t003:** Showing the spectrum of mutated genes that were found in the Swedish cohort of IRDs patients and the number of patients with pathogen variants in each specific gene.

Gene	Number of Patients
*ABCA4*	30
*AIPL1*	1
*BBS1*	1
*BBS10*	4
*BBS5*	2
*BBS9*	1
*CACNA1F*	6
*CACNA2D4*	1
*CDH23*	1
*CDH3*	1
*CDHR1*	1
*CEP290*	11
*CFAP410*	5
*CHM*	6
*CLN3*	3
*CNGB1*	1
*CNGB3*	2
*COL18A1*	4
*CRX*	3
*GUCA1A*	1
*GUCY2D*	4
*IMPDH1*	1
*IQCB1*	1
*KCNV2*	3
*KIF11*	1
*KLHL7*	1
*LRAT*	2
*MERTK*	3
*MFN2*	1
*MFRP*	1
*MYO7A*	2
*NMNAT1*	1
*NPHP1*	1
*NR2E3*	2
*NYX*	6
*OPA1*	4
*OTX2*	1
*PANK2*	1
*PCARE*	1
*PCDH15*	2
*PDE6B*	1
*PDE6C*	1
*PNPLA6*	1
*POC1B*	1
*POMGNT1*	1
*PROM1*	6
*PRPF31*	2
*PRPF8*	3
*PRPH2*	1
*RDH12*	1
*RDH5*	1
*RHO*	1
*RLBP1*	1
*RP1*	2
*RP1L1*	1
*RP2*	1
*RPE65*	1
*RPGR*	11
*RS1*	10
*TIMM8A*	1
*TRPM1*	3
*TULP1*	2
*USH1C*	1
*USH2A*	2
*WFS1*	2

## Data Availability

The authors have full control of all primary data and agree to allow the journal to review the data on request.

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
