# Peer review of "A Description of the Yield of Genetic Reinvestigation in Patients with Inherited Retinal Dystrophies and Previous Inconclusive Genetic Testing"

_genes, 2023, doi:10.3390/genes14071413_

Round 1

Reviewer 1 Report

The manuscript by Areblom et al. describes the results of genetic testing in 279 Sweden subjects with a diversity of inherited retinal dystrophies (IRD) which were tested by NGS of a panel including 322 IRD-related genes. According to the authors, no causal mutations were identified in the cohort using pre-NGS approaches. This is a nice work illustrating the benefits of NGS for precise molecular diagnosis of genetically heterogenous monogenic diseases, as IRD.

Some points to be considered by the authors are:

I suggest to include the clinical diagnosis of the cohort; for example, how many retinitis pigmentosa or Stargardt disease patients were included?

In results, the authors stated that “Pathogenic variants were found in 182 of the 279 (65%) samples”. It should be clarified if this figure is the solving rate (i.e cases with a conclusive genotype including biallelic, in trans) variants in recessive genes.

No information at all was presented on the characteristics of the pathogenic variants. Please include general information on this point as number of novel variants, identified CNVs, founder mutation effects, etc. Were the variants uploaded to Clinvar or LOVD?

“In 13 out of the 182 (7%) patients, there was a discrepancy between the diagnosis based on phenotypical or genotypical findings alone”, It would be useful to know more about these discrepancies.

Minor points

Figure 1. Showing the frequency of the different genes that were found in the study.  Please modify to “…frequency of mutated genes..”

“Table 2. Showing the spectrum of genes that were” , please modify to “spectrum of mutated genes”

The same applies for figure 2 legend.

Author Response

Some points to be considered by the authors are:

I suggest to include the clinical diagnosis of the cohort; for example, how many retinitis pigmentosa or Stargardt disease patients were included?

Response: Thanks a lot for all the valuable comments. We have now added a paragraph in the method/subject section concerning the most common clinical diagnoses found in the cohort. 94 patients suffered from retinitis pigmentosa and 24 patients had Stargardt disease.

In results, the authors stated that “Pathogenic variants were found in 182 of the 279 (65%) samples”. It should be clarified if this figure is the solving rate (i.e cases with a conclusive genotype including biallelic, in trans) variants in recessive genes.

Response: Yes, the figure is the solving rate. This has been clarified in the result section.

No information at all was presented on the characteristics of the pathogenic variants. Please include general information on this point as number of novel variants, identified CNVs, founder mutation effects, etc. Were the variants uploaded to Clinvar or LOVD?

Response: Our purpose with this study was to investigate, on a more clinical level, how efficient a broad novel NGS panel is when it comes to confirming the causative gene in previously tested but inconclusive RD-cases, but not to go into depth concerning the different pathogenic variants. Therefore, not so much facts about the variants were included originally. To clarify the characteristics of the pathogenic variants a new table, Table 2 has been added, presenting the pathogenic variants, as well as data concerning age at first examination, gender, genotype and phenotype at first examination. The variants were up-loaded to ClinVar.

“In 13 out of the 182 (7%) patients, there was a discrepancy between the diagnosis based on phenotypical or genotypical findings alone”, It would be useful to know more about these discrepancies.

                          Response: An explanation concerning this has been added to the results section.

Minor points

Figure 1. Showing the frequency of the different genes that were found in the study.  Please modify to “…frequency of mutated genes.”

                          Response: This has been corrected.

“Table 2. Showing the spectrum of genes that were” , please modify to “spectrum of mutated genes”

The same applies for figure 2 legend.

Response: This has been corrected.

Reviewer 2 Report

This is a comprehensive study involving screening a large cohort of individuals with inherited retinal dystrophies for variants in over 300 known disease genes and identifying pathogenic and likely pathogenic variants in a majority of cases.

However, a number of improvements in data presentation and level of detail should be made so that these findings are well supported by evidence and can be informative for future diagnosis of patients in a clinical setting

Specific points are as follows:

Methods:

1.     Specific algorithms and analysis pipelines used for calling coding and copy number variants should be mentioned.

Results:

1.     This study claims that ACMG class 5 and 4 variants were identified in 65% of samples. However, none of these variants are reported and evidence for pathogenicity is not provided. It is necessary to list specific variants (nucleotide and protein sequence changes) along with supporting evidence where relevant such as Minor allele frequencies in the population, pathogenicity predictions and/or segregation data. This will strengthen the paper.

2.     Phenotypes of patients in whom class 5 and 4 variants were identified are not listed. These should be listed with corresponding variants. This is especially important in cases where the authors claim discrepancy between the diagnoses by phenotypical or genotypical findings alone.

Discussion section

1.     Authors claim that the success rate was approximately the same as the general yield described for first time testing using NGS although subjects were selected unsolved cases. However, it is not clear how many patients had previously had only single-gene tests and how many had multi-gene testing. This will help better clarify the value of repeat testing in a clinical setting

Minor points

Page 4 line 118,119: ..Table 2 and … This sentence is incomplete.

The English Language in this paper is clear and requires only minor editing

Author Response

Specific points are as follows:

Methods:

Specific algorithms and analysis pipelines used for calling coding and copy number variants should be mentioned.

Response: Paragraphs concerning this has been added to the Method section.

Results:

  1. This study claims that ACMG class 5 and 4 variants were identified in 65% of samples. However, none of these variants are reported and evidence for pathogenicity is not provided. It is necessary to list specific variants (nucleotide and protein sequence changes) along with supporting evidence where relevant such as Minor allele frequencies in the population, pathogenicity predictions and/or segregation data. This will strengthen the paper.

Response: Thanks a lot for all the valuable comments. Our purpose with this study was to investigate, on a more clinical level, how efficient a broad novel NGS panel is when it comes to confirming the causative gene in previously tested but inconclusive RD-cases, but not to go into depth concerning the different pathogenic variants. Therefore, not so much facts about the variants were included originally.

To clarify the characteristics of the pathogenic variants a new table, Table 2 has been added, presenting the pathogenic variants, as well as data concerning age at first examination, gender, genotype, and phenotype at first examination. In the section concerning the methods there is also an explanation of how the variants were interpreted. (Variant data was annotated using a collection of tools (VcfAnno and VEP) with a variety of public variant databases including but not limited to gnomAD, ClinVar and HGMD).

  1. Phenotypes of patients in whom class 5 and 4 variants were identified are not listed. These should be listed with corresponding variants. This is especially important in cases where the authors claim discrepancy between the diagnoses by phenotypical or genotypical findings alone.

Response: The new table, Table 2, includes both the genotype (with the pathogenic variants) and the phenotype. A paragraph explaining the discrepancies has also been added to the Result section.

Discussion section

1 Authors claim that the success rate was approximately the same as the general yield described for first time testing using NGS although subjects were selected unsolved cases. However, it is not clear how many patients had previously had only single-gene tests and how many had multi-gene testing. This will help better clarify the value of repeat testing in a clinical setting.

Response: 157 of the patients had had multi-gene testing previously, while the remaining 122 had had single genes tested. However, in the majority of these cases, DNA was sent to several different laboratories that provided analyses of different genes or set of genes. WES was made as an extra analysis in just a few cases. A paragraph concerning this has been added to the method section.

Minor points

Page 4 line 118,119: ..Table 2 and … This sentence is incomplete.

Response: This has been corrected.

Reviewer 3 Report

Areblom and co-authors present their data in genetic re-evaluation of 279 patients clinically diagnosed with an IRD but without genotypic diagnosis. The authors indicate that of the 279 patients they are able to provide genotypic diagnosis in 182 using a NGS IRD panel targeting 322 loci. They indicate they are basing it on the Blueprint genetics panel (which covers more genes at 351). They identify profiles of which genes are most prevalent after re-testing and discuss how this is related to globally identified data.

Overall this study has major flaws. First of all it is unclear what previous testing these patients had. It is not surprising that a person with single gene testing is now provided with a genetic diagnosis now that a large NGS panel is used. Also it is unclear when the testing was done in each case. A test done in 1995 when the study began looking at the data would be inferiorly given the advancement in NGS technology in the last few decades. There is no breakdown given on the 279 patients they chose as to when each had a test and what test was done. 

The methods section is also severely lacking. Why did the authors chose to do a NGS panel of 322 genes and not all 351 that is covered in the Blueprint panel? What methodology did they use to make DNA libraries from isolated blood? They indicate the did assessment of non-coding and copy number variants but what metrics do we have that they did this? Also what variants did they identify in their re-analysis? Were any of them non-coding or CNVs to justify re-testing with their NGS panel? My biggest concern is why did they just not pursue testing through Blueprint if all they were doing is just using their panel?

The Subject data in the methods is also not adequate. They list subjects had previous testing, but we have no idea to what extent and when this testing was done. Also they picked 279 patients with inconclusive testing, but what does that mean? They do not explain if mean no variants identified or one variant (monoallelic) in a gene known to cause autosomal recessive disease. Taking that into account was their solve rate higher in monoallelic cases where they already had an idea of the gene? Were any of these in non-coding regions which are hypothesized to explain some cases of missing heritability?

Finally, they list that 7% did not match the genotypic diagnosis and this suggests to me the examinations were not as detailed as expected. For example, if ERG testing was done in all cases, then the case they highlight with CACNA1F should have highlighted characteristic B-wave findings to raise their suspicion for CSNB. 

Their discussion points of how their identified results differ from other studies is difficult to follow. The biggest takeway should be they worked with a mainly pediatric and adolescent population where many of the others were mixed or adult only to explain the differences, which they do not make clear.

Overall, this study highlights results from a single site and shows NGS testing can help identify genetic diagnosis in IRDs, but this work does not advance our understanding. This is more of a case report of findings than a true scientific study. In order to be considered for publication a more rigorous paper would be expected to put forth.

Minor corrections, overall decent writing

Author Response

Overall this study has major flaws. First of all it is unclear what previous testing these patients had. It is not surprising that a person with single gene testing is now provided with a genetic diagnosis now that a large NGS panel is used. Also it is unclear when the testing was done in each case. A test done in 1995 when the study began looking at the data would be inferiorly given the advancement in NGS technology in the last few decades. There is no breakdown given on the 279 patients they chose as to when each had a test and what test was done. 

Response: Thanks a lot for the valuable comments. We agree that more detailed information concerning the previous tests for each individual could have been beneficial, but that would have included very much information, risking to be incalculable and the main purpose of this study was not to exactly compare previous and current test methods but to find out to what extent renewed DNA-analysis with a widely accessible and decently affordable method as a broad NGS panel can help us to solve the genotype in a cohort of patients previously tested with a mix of different methods such as the real life setting in our clinics.

To clarify the status concerning previous testing to some extent, we have added a paragraph to the section Material and Methods/ subjects, where we specify how many patients that were investigated with single gene tests (122) and how many patients that had been investigated with APEX or NGS panels (157).

157 of the patients had had multi-gene testing previously, while the remaining 122 had had single genes tested. However, in the majority of these cases, DNA was sent to several different laboratories that provided analyses of different genes or set of genes. WES was made as an extra analysis in just a few cases.

The methods section is also severely lacking. Why did the authors chose to do a NGS panel of 322 genes and not all 351 that is covered in the Blueprint panel? What methodology did they use to make DNA libraries from isolated blood? They indicate the did assessment of non-coding and copy number variants but what metrics do we have that they did this? Also what variants did they identify in their re-analysis? Were any of them non-coding or CNVs to justify re-testing with their NGS panel? My biggest concern is why did they just not pursue testing through Blueprint if all they were doing is just using their panel?

Response: The testing was done through Blueprint Genetics (which is mentioned in the method section) and the reason for why the panel included only 322 genes is that the analyses were performed in 2021 and by that time, the NGS retinal dystrophy panel included those 322 genes instead of the 351 that are investigated today. (It has taken some time to compile the material and write the manuscript etc).

To clarify the methods, we have added two paragraphs to the Material and Methods section including more information concerning the NGS analysis, analysis of non-coding regions and copy number variations as well as the variant interpretation. Moreover, a section concerning DNA extraction has been added.

We have also made an extra table, Table 2, presenting genotype including the pathogenic variant, phenotype as well as demographic data of the patients.

The Subject data in the methods is also not adequate. They list subjects had previous testing, but we have no idea to what extent and when this testing was done.

Response: We agree that more detailed information concerning the previous tests for each individual could have been beneficial, but that would have included very much information, risking to be incalculable and the main purpose of this study was not to exactly compare previous and current test methods but to find out to what extent renewed DNA-analysis with a widely accessible and decently affordable method as a broad NGS panel can help us to solve the genotype in a cohort of patients previously tested with a mix of different methods such as the real life setting in our clinics.

To clarify the status concerning previous testing to some extent, we have added a paragraph to the section Material and Methods/ subjects, where we specify how many patients that were investigated with single gene tests (122) and how many patients that had been investigated with APEX or NGS panels (157).

157 of the patients had had multi-gene testing previously, while the remaining 122 had had single genes tested. However, in the majority of these cases, DNA was sent to several different laboratories that provided analyses of different genes or set of genes. WES was made as an extra analysis in just a few cases.

Also they picked 279 patients with inconclusive testing, but what does that mean? They do not explain if mean no variants identified or one variant (monoallelic) in a gene known to cause autosomal recessive disease. Taking that into account was their solve rate higher in monoallelic cases where they already had an idea of the gene? Were any of these in non-coding regions which are hypothesized to explain some cases of missing heritability?

Response: With “inconclusive previous DNA test results” we mean that either no pathogenic variant at all had been identified or that only one pathogenic variant had been identified in genes known to cause autosomal recessive disease. An explanation concerning this has been added to the Materials and Methods/Subjects section.

The added, Table 2, includes all the pathogenic variants clarifying their types and locations.

Finally, they list that 7% did not match the genotypic diagnosis and this suggests to me the examinations were not as detailed as expected. For example, if ERG testing was done in all cases, then the case they highlight with CACNA1F should have highlighted characteristic B-wave findings to raise their suspicion for CSNB.

Response: Of course, the typical ERG pattern with electronegativity should raise the suspicion of CSNB, but in the era before OCT those patients could be mixed up with XLRS in a child who was not too eager to sit still during fundus examination. Thus, some CSNB cases were mistaken for XLRS and also one case of CSNB was mistaken for choroideremia. Moreover, some early cases of choroideremia were mistaken for RP. This has been clarified in the Result section and further emphasizes the importance of a careful investigation of both genotype and phenotype, which is also one of the points in the discussion.

Their discussion points of how their identified results differ from other studies is difficult to follow. The biggest takeway should be they worked with a mainly pediatric and adolescent population where many of the others were mixed or adult only to explain the differences, which they do not make clear.

Response: The discussion concerning the genetic spectrum in this Swedish cohort is mainly carried out since this is, to our knowledge, the first study mapping the genetic spectrum in a Swedish cohort and thus it is interesting to note similarities and differences to other geographic areas and populations. This can be of importance when it comes to national initiatives concerning research regarding gene-based therapies. For example, in Sweden and Finland, we don´t have many patients with RPE65 associated retinal dystrophies and thus have less use of Voretigene Neparvovec.

A sentence explaining that this is the first larger study on the genetic spectrum of IRDs in a Swedish cohort has been added to the discussion to clarify why the comparison to cohorts in other regions are carried out.

Overall, this study highlights results from a single site and shows NGS testing can help identify genetic diagnosis in IRDs, but this work does not advance our understanding. This is more of a case report of findings than a true scientific study. In order to be considered for publication a more rigorous paper would be expected to put forth.

Response: We hope that the above-mentioned updates with extended sections concerning DNA-testing and DNA extraction in Material and Methods and the new Table 2 with thorough data concerning the pathogenic variants and phenotype as well as other clarifications in the text will improve our manuscript so it can be considered for publication.

Round 2

Reviewer 1 Report

The authors have answered all queries from this reviewer, thank you. The manuscript has been improved very nicely.

Author Response

The authors have answered all queries from this reviewer, thank you. The manuscript has been improved very nicely.

Response: Thanks for the previous useful comments and for the approval.

Reviewer 3 Report

The revisions to the initial manuscript do not change the message of this article, unfortunately. As a study the title is very misleading suggesting it is a genetic reinvestigation because the authors provide no information of previous genetic results. In the revision they list that 122 subjects had single gene testing (actually this is also inaccurate as the authors themselves say they had a range of genes of several occasions so what is it?) and 157 NGS panels but no mention of what those results are. One cannot call it a re-investigation if we do not know what the initial investigation yielded.

They clarify inconclusive tests mean no variant or monoallelic variant, however what were these variants? As was raised in the initial review comments: “Taking that into account was their solve rate higher in monoallelic cases where they already had an idea of the gene? Were any of these in non-coding regions which are hypothesized to explain some cases of missing heritability?”. The Blueprint is limited in non-coding variant detection as it looks at a 20 bp overhang and then select number of known variants (mostly in ABCA4).

The authors inserted Table 2 to report the variants they identified, but without knowing what the initial study showed, this is not a re-investigation. For example in the cases with 2 heterozygous variants or 2 homozygous variants in autosomal recessive disease, why were they not found on initial testing. For example in ABCA4-associated Stargardt that has a clear phenotype was ABCA4 not tested and if so why were these variants initially missed? This is the central question that was raised in the initial review and the authors did not address it again in the revision.

The authors had a lot of text in the methods but this is superfluous since this was all done by a commercial lab (Blueprint Genetics). The methods seem to be just taken from the company as it was not seemingly done by the authors. Did the authors do any of the DNA extraction and genetic analysis or was this all done by Blueprint and they simply are reporting what the output report from Blueprint told them?

Essentially they took a patient cohort and report a presumed genetic diagnosis of 65% which is what would expect from NGS testing on various larger scale studies across different populations. We can also only presume this is the case as familial segregation was not done to confirm variants were in trans. 

None

Author Response

We thank the reviewer for all comments regarding our manuscript and we are sorry that our main purpose has not been clear.

Since about 1990, our main ambition in the clinic has been to carefully verify the phenotype in every patient to provide reliable information regarding visual handicap and prognosis, particularly in young patients with suspected inherited retinal dystrophy.  Clinical methods have developed over the years and today, beyond standard clinical examination, we can examine also in young children and handicapped patients with full-field ERG, multifocal ERG, and OCT during general anesthesia to evaluate the phenotype thoroughly. Over the years, we have also strived to solve the genotype in every case, but not always succeeded. Therefore, the purpose of this study was to evaluate if a commercially available broad NGS panel from a certified genetic laboratory (available and maybe also affordable in many countries) could be of use to solve the genotype/phenotype correlation in  group of young patient (< 25 years) previously examined regarding the phenotype but in whom the genotype was still unsolved.

The revisions to the initial manuscript do not change the message of this article, unfortunately. As a study the title is very misleading suggesting it is a genetic reinvestigation because the authors provide no information of previous genetic results. In the revision they list that 122 subjects had single gene testing (actually this is also inaccurate as the authors themselves say they had a range of genes of several occasions so what is it?) and 157 NGS panels but no mention of what those results are. One cannot call it a re-investigation if we do not know what the initial investigation yielded

They clarify inconclusive tests mean no variant or monoallelic variant, however what were these variants? As was raised in the initial review comments: “Taking that into account was their solve rate higher in monoallelic cases where they already had an idea of the gene? Were any of these in non-coding regions which are hypothesized to explain some cases of missing heritability?”. The Blueprint is limited in non-coding variant detection as it looks at a 20 bp overhang and then select number of known variants (mostly in ABCA4).

Response: In this study, we have chosen to focus on the evaluation of the possible benefit of renewed DNA-testing with a commercially available broad NGS panel from a certified genetic laboratory (available and maybe also affordable in many countries) in unsolved cases and therefore we have not gone into depth concerning previous inconclusive DNA test results, but concentrated on the outcome of the NGS analysis in relation to the phenotypes evaluated during the years.

Equally, our main goal was not to further interpret the results from a certified laboratory, but to investigate the yield of confirmed genotypes, even though we continuously follow the discussions about the interpretation of different genotypes in several publications and agree with the reviewer that is an important issue.

The authors inserted Table 2 to report the variants they identified, but without knowing what the initial study showed, this is not a re-investigation. For example in the cases with 2 heterozygous variants or 2 homozygous variants in autosomal recessive disease, why were they not found on initial testing. For example in ABCA4-associated Stargardt that has a clear phenotype was ABCA4 not tested and if so why were these variants initially missed? This is the central question that was raised in the initial review and the authors did not address it again in the revision

Response: As far as we understand, the results of our study show that modern comprehensive genetic testing is continuously improved by time including new pathogenic genetic variants and thus it is of great benefit to test patients with unsolved genotype again after some years. At least in Sweden, there has been a lot of discussions among colleagues and heads of nationa health service if this is of use and now, we think that our study can support the benefit of repeated testing after some years.

The authors had a lot of text in the methods but this is superfluous since this was all done by a commercial lab (Blueprint Genetics). The methods seem to be just taken from the company as it was not seemingly done by the authors. Did the authors do any of the DNA extraction and genetic analysis or was this all done by Blueprint and they simply are reporting what the output report from Blueprint told them?

Response: We agree that the text regarding methods could be shorter, but we have been asked by reviewers to include the data. During these 30 years, DNA extraction has been carried out in house in cooperation with the department of Clinal Chemistry at Skåne University hospital in Lund.

When it comes to the genetic analysis, the purpose of this study was to evaluate if a commercially available broad NGS panel from a certified genetic laboratory (available and maybe also affordable in many countries) could be of use to solve the genotype in previously unsolved cases, therefore all the DNA testing was done by Blueprint Genetics.

Essentially, they took a patient cohort and report a presumed genetic diagnosis of 65% which is what would expect from NGS testing on various larger scale studies across different populations.

Response: Our point here is that it is useful also to repeat genetic testing in unsolved cases after some years. E.g, the positive yield of testing for the patients with the shortest re-test interval (previously tested between 2016-2020 with APEX – or NGS panels) also was 65%, which means a similar positive success rate as for the whole group, indicating the usefulness of re-testing with quite short interval.

Moreover, we report the genotypic spectrum in a Swedish cohort which, to our knowledge has not been published before.

We can also only presume this is the case as familial segregation was not done to confirm variants were in trans.

Response: We agree that it would be ideal with segregation analysis in all cases with compound heterozygosity, but in this study, it was not possible to get in touch with and test relatives to all of the patients. In many cases, Blueprint Genetics could confirm, from the NGS data, that the variants were in trans and in all other cases with compound heterozygosity we were very careful in the interpretation and considered the genotype as causative only if it was fully consistent with the phenotype. A paragraph commenting on this has been added to the discussion.